# Low Intensity, Transcranial, Alternating Current Stimulation Reduces Migraine Attack Burden in a Home Application Set-Up: A Double-Blinded, Randomized Feasibility Study

**DOI:** 10.3390/brainsci10110888

**Published:** 2020-11-21

**Authors:** Andrea Antal, Rebecca Bischoff, Caspar Stephani, Dirk Czesnik, Florian Klinker, Charles Timäus, Leila Chaieb, Walter Paulus

**Affiliations:** Department of Clinical Neurophysiology, University Medical Center, Georg-August University, 37075 Göttingen, Germany; rebecca.bischoff@stud.uni-goettingen.de (R.B.); Cstephani@med.uni-goettingen.de (C.S.); dczesnik@gwgd.de (D.C.); fklinker@med.uni-goettingen.de (F.K.); ctimeus@med.uni-goettingen.de (C.T.); Leila.chaieb@med.uni-goettingen.de (L.C.); wpaulus@med.uni-goettingen.de (W.P.)

**Keywords:** tACS, migraine, acute treatment, visual cortex, transcranial stimulation

## Abstract

**Background:** Low intensity, high-frequency transcranial alternating current stimulation (tACS) applied over the motor cortex decreases the amplitude of motor evoked potentials. This double-blind, placebo-controlled parallel group study aimed to test the efficacy of this method for acute management of migraines. **Methods:** The patients received either active (0.4 mA, 140 Hz) or sham stimulation for 15 min over the visual cortex with the number of terminated attacks two hours post-stimulation as the primary endpoint, as a home therapy option. They were advised to treat a maximum of five migraine attacks over the course of six weeks. **Results:** From forty patients, twenty-five completed the study, sixteen in the active and nine in the sham group with a total of 102 treated migraine attacks. The percentage of terminated migraine attacks not requiring acute rescue medication was significantly higher in the active (21.5%) than in the sham group (0%), and the perceived pain after active stimulation was significantly less for 2–4 h post-stimulation than after sham stimulation. **Conclusion:** tACS over the visual cortex has the potential to terminate migraine attacks. Nevertheless, the high drop-out rate due to compliance problems suggests that this method is impeded by its complexity and time-consuming setup.

## 1. Introduction

Transcranial magnetic (TMS) and direct current stimulation (tDCS) applied over the visual or motor areas have shown efficacy in the acute and prophylactic treatment of migraines in placebo-controlled studies [1,2,3,4,5,6,7,8,9,10,11,12] (for a recent meta-analyses see [13]). The application of two-pulses of TMS over the visual cortex or over the painful area has been claimed to ameliorate or terminate migraine pain [3,5]. This effect is assumed to be based on influencing neuronal activity and, in the case of an aura, interfering with the occurrence of cortical spreading depression in the early phase of the migraine attack [14]. 

In healthy subjects, transcranial alternating current stimulation (tACS) with 0.4 mA at 140 Hz applied over the primary motor cortex (M1) can significantly decrease the amplitude of motor evoked potentials (MEPs) at rest [15]. In the present study, we aimed to target the visual cortex of migraine patients at the onset of the migraine attack by having the patient apply tACS at home. We applied this kind of “inhibitory” stimulation based on the results of previous studies, suggesting that the migraine is associated with higher visual neuronal excitability or responsiveness (e.g., [16,17,18,19,20,21,22,23]). Although there are no studies in which this kind of stimulation was applied over the visual cortex, we hypothesized that modifying cortical activity through the application of high-frequency transcranial oscillations might adjust behaviorally “maladaptive” brain states and induce a new balance, forcing the network to restore adequate synchronization and excitation/inhibition balance. 

Transcranial stimulation, including tACS, is normally administered by medical professionals in a clinical setting to ensure correct administration of the treatment. The necessity to visit the hospital immediately to treat a migraine attack makes this type of treatment unpractical. The necessity of repeated visits may also increase drop-out rates in long-term studies, e.g., in depression [24], or even interfere with patient-recruitment. Self-administration of tACS by the patients or with the help of their relatives would counteract this disadvantage. The feasibility of this approach, using tDCS, has been demonstrated for several disorders, including depression, Parkinson’s disease, Alzheimer’s disease, trigeminal neuralgia, and menstrual migraine [25,26,27,28]. Furthermore, home stimulation can significantly reduce personal costs and more importantly, the therapy of the patients can be continued and remotely supervised even during a pandemic [29]. 

The present study is aimed at promoting a safe and feasible protocol for self-administered tACS in the home therapy of migraine attacks. This protocol has not been used in patients before. Special attention was paid to optimal user training for a maximally standardized and reproducible transcranial stimulation setup. 

## 2. Methods

All aspects of this study conformed to the Declaration of Helsinki; written informed consent was given by all study participants. The experimental protocol was approved by the ethics committee of the Medical Faculty of the University of Göttingen (code: 1/5/03, amendment: 19.04.12).

### 2.1. Patients

Forty migraine patients were recruited from outpatient clinics and private practices for the study. This estimation was based on the sample size of previous feasibility studies, treating acute migraine attacks with brain stimulation methods (for a review see [13]). At this stage, we aimed to prove the feasibility of the methodology. Inclusion criteria were migraine with or without aura and disease duration ≥6 months [30]. Exclusion criteria were significant chronic health disorders, diagnosed neuropsychiatric disorders, pregnancy or breast feeding, history of substance abuse or dependence, a history of neurological disorders other than migraine, an implanted pacemaker and cranial metallic hardware. All patients were naïve to transcranial stimulation and none took prophylactic migraine medication during the study period. If applicable, female patients were advised to continue contraception (that was started at least 6 months prior to enrollment into the study) during the whole study period. None of the patients had a history of acute migraine medication overuse. 

### 2.2. Experimental Design

The primary endpoint of this double-blind, placebo-controlled study was the termination of the migraine attacks within two hours post-stimulation (numerical analogue scale (NAS) values <1). If the pain after this period was still present and cannot be tolerated by the patients, the patients were allowed to take their regular acute migraine medications. 

Patients were asked to maintain a headache diary throughout the study duration. During the study, the frequency of the migraine attacks was recorded, including onset and duration of the pain, number of migraine-related days and the type of analgesics taken. Patients were advised to document the degree of pain on a NAS with severity ratings ranging from 0 to 10 at onset of a migraine attack as well as 1 h and 2, 4, 8, 24, and 48 h thereafter. A NAS is frequently used as a valid and reliable measurement of migraine pain [31]. 

### 2.3. Transcranial Alternating Current Stimulation

The patients were assigned to receive either treatment “A”, referring to real or “B”, referring to sham stimulation, according to a computer randomization list. The battery-driven stimulators (NeuroConn, Ilmenau, Germany) were preprogrammed and coded by the coordinating investigator, who had no contact with the patients. During programming, the type of the stimulation can be saved; however, during stimulation, no differences between real and sham stimulations on the screen can be detected. Unknown to the patients, the parameters used during the home stimulation sessions, including the time and duration of the sessions, were stored in the stimulator.

The stimulation was then applied by the patient at home. Since electrode preparation and positioning are essential factors in reproducible remotely-supervised treatment [32], the patients were given detailed instructions and a training session in the department before being allowed to use the stimulator. Saline-soaked sponge electrodes were used. The stimulating electrode (4 × 4 cm) was placed over the Oz and the return electrode (5 × 7 cm) over the Cz electrode positions and fixed with the elastic band. This was done by the patients, without help. 

According to a modeling study, these electrode positions present current densities in the range of 0.05–0.15 A/m^2^, the higher intensities being allocated to the medial, as compared to the lateral occipital cortex [33]. tACS with 0.4 mA was applied for 15 min, including 20 s ramp-up and ramp-down phases. For sham stimulation, the electrodes were placed in the same positions as for active stimulation, but the stimulator was turned off automatically after 30 s of stimulation. Both the patients and the training investigator were blinded with regard to the type of tACS applied. The patients were instructed to start the stimulation session at the beginning of the migraine attack (e.g., after the appearance of aura or pain). The patients were aware of the fact that they would receive either sham or real stimulation.

Since any potential adverse effects (AEs) of this technique in a patient population are not yet known, the patients were asked to report AEs during the whole study period and they were instructed on what do to do in case of the occurrence of severe AEs. Furthermore, they completed a questionnaire after the whole stimulation session. The questionnaire contained rating scales for the presence of discomforting sensations such as pain, tingling, itching or burning under the electrodes due to tACS [34] (1 = very mild and 5 = extremely strong intensity). 

### 2.4. Statistical Analysis

Repeated measures ANOVA was used to test for differences in pain perception with the factors “type of stimulation” (active and sham) and “time” (before and after treatment, hours). Mann–Whitney U test was used to compare the number of terminated attacks (with and without medication) in the active and sham groups. With regard to the primary endpoint, a *p*-value of ≤0.05 was considered significant. All other analyses are considered exploratory and confidence intervals as well as *p*-values are reported without correction for multiple testing. 

The incidences of AEs were coded in a binary system (no = 0, yes = 1) and the severities of the AEs were rated using a NAS from one to five, one being very mild and five being of an extremely strong intensity of any given AE. 

## 3. Results

Forty patients were randomized using a computer algorithm to get real (25 patients) or sham (15 patients) stimulation. Fifteen patients, nine from the active and six from the sham group, had to be excluded from evaluation during the course of the study. Eight were excluded because they failed to perform any stimulation at home. Four of these were only identified by analyzing the stimulator memory. Four patients had no migraine attacks during the study period, two patients decided to withdraw without giving any reason and one patient experienced a panic attack before the stimulation. Therefore, only twenty-five patients returned a valid migraine diary, the demographical characteristics and medical history of which are summarized in Table 1. The demographical characteristics of the patients related to the disease (duration of the disease and number of attacks/year) did not differ significantly between the active and sham stimulation groups (t-test, *p* > 0.1).

These 25 patients suffered a total of 102 documented migraine attacks during the study: 65 migraine attacks were treated in the active group (mean: 4.06 attacks/patient, range: 1–5) and 37 in the sham group (mean = 4.11 attacks/patient, range 1–5). In the active group, 27 attacks were treated with drugs within two hours after the stimulation compared to 14 in the sham group (41.5% vs. 37.8%) (Figure 1).

During the attacks without pharmacological interventions, the pain abated within two hours post-stimulation in 14 of the 38 attacks in the active group, but in none of the 23 attacks in the sham group, showing a statistical significant difference between the two groups (*p* < 0.001). If we consider the pain severity in both groups, it was significantly lower after tACS than after sham stimulation in the first two–four hours (main effect: F(1,35) = 9.173, *p* < 0.0045; interaction: F(7,245) = 6.62, *p* < 0.00001) (Figure 2). According to the pain diaries, none of the documented attacks reoccurred after 24 and 48 h.

With regard to the presence of aura, it was not possible to make reliable statistical analysis because of the low number of patients in each group, however, no different effects were observed in patients with or without aura.

AEs of tACS: No medical interventions other than acute migraine medications were required; 23 patients completed the questionnaire, 15 in the active and 8 in the sham group. Table 2 summarizes the AEs due to stimulation.

## 4. Discussion

Our hypothesis that “inhibitory” tACS over the visual cortex could be an effective acute treatment option was based on data suggesting that migraine is associated with higher neuronal excitability or responsiveness (e.g., [16,17,18,19,20,21,22,23]) and the observation that 0.4 mA 140 Hz tACS over the motor cortex probably decreases cortical excitability [15]. Accordingly, we found that a significantly higher percentage of migraine attacks were terminated within two hours post-stimulation in the tACS group. Nevertheless, only less than one in four of the attacks could be completely terminated by this intervention; in almost half of the attacks additional acute medication was required. In the sham group, 38% of the attacks were treated with drugs and none of them responded to the sham stimulation.

Despite the variety of pharmaceutical options available for the prophylaxis or acute treatment of migraines, a substantial proportion of patients remains resistant to drug therapy. The efficacy response rates for these therapies range around 40–50% in most studies, suggesting that responsive patients generally represent less than half of the population [35,36,37]. Several non-pharmaceutical alternatives, such as exercise and acupuncture, have been compared with common prophylactic medications [38,39,40] and seem to offer some benefit for migraine patients [41]. Non-invasive neuromodulation, including magnetic and low intensity electric stimulation, is an emerging treatment strategy for migraine headache disorders. These methods provide the distinctive opportunity of avoiding disparate medication AEs and interactions. Two pulses of TMS at low or high intensity were applied in an open-label study during acute migraine attacks. Stimulation was over the region of pain in patients without aura, or over the visual cortex in patients with aura [5]. Pain intensity was reduced by 75% up to 20 min post-TMS. Furthermore, 32% of patients reported no further headache for up to 24 h after one treatment, 29% after two treatments and 40% after three treatments. In another study, 164 patients with aura were stimulated over the visual cortex within one hour of aura onset using a randomized, double-blind, parallel-group, sham-controlled design [3]. Up to three attacks were treated over a three-month period. Real TMS was more effective than sham stimulation in alleviating pain at two hours (39% vs. 22%), and for sustained pain relief at 24 h (29% vs. 16%) and 48 h (27% vs. 13%). Based on telephone interviews, single pulse TMS in 190 episodic or chronic migraine patients reduced the number of headache days after 12 weeks of treatment in nearly 60% of patients in whom acute medications were contraindicated or ineffective [42]. Nevertheless, the discontinuation rate was 55% in this study. Repetitive TMS (rTMS) as a preventative treatment both for episodic and chronic migraine has resulted in mixed outcomes [4,43,44]. Generally, rTMS is promising with moderate evidence in acute and prophylactic treatment that it contributes to reductions in headache frequency, duration, intensity, abortive medication use, depression, and functional impairment compared to baseline, when the M1 or the frontal cortex were stimulated, using “excitatory” frequencies [45]. Nevertheless, many of the studies reported non-significant changes compared to sham treatment. 

So far, the efficacy of prophylactic anodal and cathodal tDCS has been primarily tested with diverse results, mainly due to the different setting procedure and location of electrodes [1,6,10,46,47]. Nevertheless, in different studies, it has been observed that the anodal stimulation of the primary motor cortex and the cathodal stimulation of the occipital cortex are associated with a significant reduction in the number of headache days, consumption of tablets, and pain intensity, and a significant increase in the number of headache-free days [48].

To our knowledge, tACS has never been employed before in patients with migraine. Previous data suggest that stimulating the motor cortex of healthy young subjects with 140 Hz tACS at 0.4 mA can decrease the amplitude of MEPs [15] for more than one hour after stimulation. However, it is not clear what is the exact neuronal underlying mechanism. It is hypothesized that 140 Hz (at this lower intensity) only facilitated intracortical inhibitory networks of corticospinal neurons and may have inhibited intracortical facilitatory influences on corticospinal neurons. Using lower frequencies in the alpha range and higher intensities, tACS induced increased alpha power [49,50]. We assume that tACS over the visual cortex may not only reduce local excitability but possibly modify the activity of the brainstem through nociceptive pathways [30]. It is suggested that there is a functional connection between the visual cortex and brainstem second-order nociceptors in the spinal trigeminal nucleus. Therefore, inhibiting the projection from the visual cortex to the brainstem might result in less pain during attacks. With regard to the stimulation montage, the placement of return electrode over the motor cortex that was achieved in previous tDCS studies [46], and based on the fact that 140 Hz stimulation effectively modified the size of MEP amplitudes [15], might result in better clinical efficacy. 

Home therapy was well tolerated by the patients who used the stimulator. The majority of user feedbacks after stimulation concerning the efficacy was either positive or neutral. Nevertheless, the main reason for the substantial fraction of non-compliance might be the time-consuming task of positioning the electrodes before stimulation as compared to taking a pill. This is a general indicator for a problematic feasibility of this kind of intervention. In future studies, family members should be involved in the training sessions, when stimulation is to be performed at home as an acute intervention. In addition to this, although the patients were instructed to start the stimulation immediately after the first signs of the migraine attack appeared, many of them probably did not do that, reflected by their relatively high baseline NAS values. Indeed, if the attack starts at work or in other situations other than at home, the stimulation cannot be started immediately. Furthermore, due to the high drop-out rate, the current study is limited by the small remaining sample size. 

## 5. Conclusions

In summary, acute application of tACS over the visual cortex (0.4 mA, 140 Hz) for 15 min was able to terminate migraine attacks. Despite home treatment, the logistic effort was high with strict training and supervision by healthcare professionals. Improved strategies to further simplify the procedure will certainly reduce the drop-out rate. Fine tuning of dose titration may also increase efficacy. Furthermore, strategies to increase efficacy in combination of neuroplasticity modification with migraine prophylactic drugs warrant further investigations.

## Figures and Tables

**Figure 1 brainsci-10-00888-f001:**
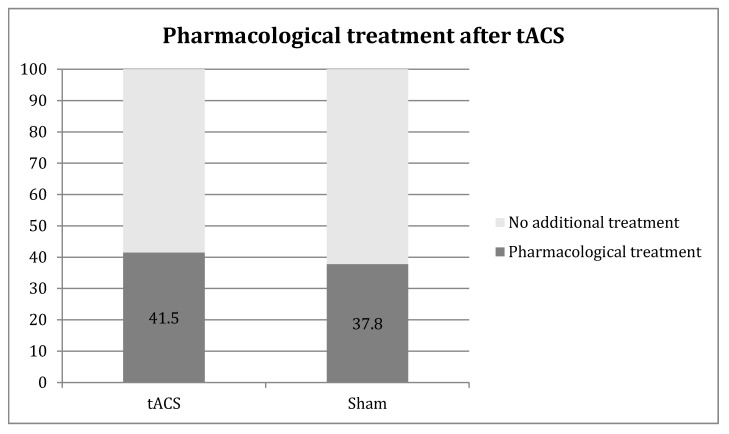
Medication use after transcranial alternating current stimulation (tACS) treatment. The Y axis represents 100% of the migraine attacks.

**Figure 2 brainsci-10-00888-f002:**
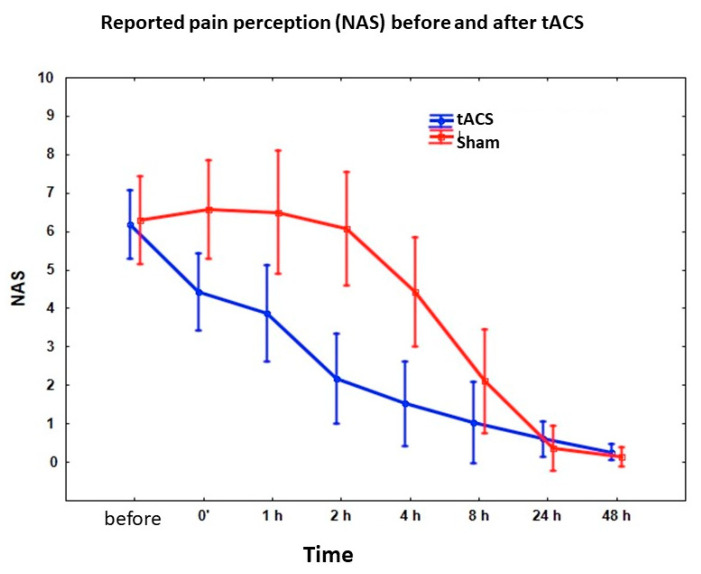
Effect of 140 Hz 0.4 mA tACS applied over the visual cortex on pain severity during migraine attack. X axis represents numerical analogue scale (NAS) values before and after stimulations. Bars represent 95% confidence intervals.

**Table 1 brainsci-10-00888-t001:** Demographics and medical history of the patients.

	tACS (*n* = 16)	Sham (*n* = 9)
**With aura**	9	5
**Without aura**	7	4
**Mean age (SD)**	31.1 (8.9)	28.1 (10.5)
**Mean duration in years (SD)**	13.7 (7.8)	14.8 (10.3)
**Mean number of attacks/year (SD)**	28.7 (18.5)	42.8 (42.2)
**Pain localization**		
unilateral	11	5
bilateral	5	4
**with Family history**	9	7
**Medication**		
Acetylsalicylic acid (Aspirin)	2	1
Triptans	4	3
Ibuprofen	2	1
Paracetamol	4	3
Others		
-Antidepressants	2	0
-Metamizole	1	0
-Thyroid Hormone	1	0
**Oral contraception**	9	5
**Smokers**	4	1

**Table 2 brainsci-10-00888-t002:** Adverse effects of tACS reported after stimulation. N: number of patients; MI = mean intensity (1 = very mild and 5 = extremely strong intensity).

	Pain under the Electrodes		Tingling		Itching	
	N	MI	N	MI	N	MI
**tACS (*n* = 16)**	1	2	5	1.8	5	1.4
**Sham (*n* = 9)**	1	3	4	1.8	2	1.3
	**Nervousness**		**Fatigue**		**Unpleasantness**	
	N	MI	N	MI	N	MI
**tACS (*n* = 16)**	1	4.0	6	2.2	2	2.0
**Sham (*n* = 9)**	3	2.5	6	2.2	5	3.3

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
