# Peer review of "Low Intensity, Transcranial, Alternating Current Stimulation Reduces Migraine Attack Burden in a Home Application Set-Up: A Double-Blinded, Randomized Feasibility Study"

_brainsci, 2020, doi:10.3390/brainsci10110888_

Round 1

Reviewer 1 Report

In the study by Antal et al. they examined the use of 15 minutes of 4 mA, 140 Hz tACS over the visual cortex on the ability to terminate migraine attacks (2-hours post-stimulation) and pain due to migraines. Their results showed that pain was abated in 14 of 38 attacks in the active group compared to 0 of 23 in the sham group. Furthermore, they showed that self-reported pain was significantly less in active compared to sham group up to 4 hours post-stimulation.

The study is on an interesting topic with clear clinical applications that could benefit many people. The small sample size limits the application of the study’s findings, but I believe it still warrants publication after addressing some matters.

Line 64 – 40 migraine patients estimation based on sample size calculations – was this calculated per group? Only 25 (16 real vs. 9 sham) total completed the study, so this would appear to be a severely under-powered study based on your sample size calculations.

Line 84 – has VAS shown to be valid and reliable for migraine pain? Please provide a reference

Line 103-105 – since the stimulator was turned off after 30 seconds of stimulation, would the participants not be aware of this sudden offset of the machine rendering the “blinding” irrelevant?

Lines 116-117 – Why was a Chi-squared test used for this analysis? Chi-squared test are used to show whether there is a relationship between two variables (without any direction of the relationship), and I don’t believe this is what you were trying to achieve with this analysis. An independent samples t-test or more likely Wilcoxon-Mann-Whitney (since it unlikely normally distributed data) should be used instead. Adjust results accordingly.

Table 1- The sham group reported a lot more mean number of attacks per year – did you perform statistical analysis on this? If not, you should since it could be argued that the sham group suffered from a more serious migraines.

Lines 137 – 138 – You should report average per person rather than total migraine attacks due to the uneven size of groups.

Lines 143 – 144 & Figure 2 – The active vs. sham looks different at 8 hours (more so than 0’); please confirm that pain was less in active at 0’ but not 8 hours.

Line 145 – spelling mistake - none not non

Table 2 – please fix font size and bolding.

Line 210 - Drop-out/exclusion rate was high suggesting at home application, although in essence potentially effective, does not necessarily ameliorate adherence to treatment due to barriers of attending a clinic. I think the claim that home therapy was well tolerated may be too general as those that didn’t use it and excluded from study/dropped-out may have not tolerated, hence not completed stimulation. Please reflect this in your discussion.

Figure 1 – Please define that VAS is used for pain severity in both title and y-axis.

Author Response

Reviewer 1.

In the study by Antal et al. they examined the use of 15 minutes of 4 mA, 140 Hz tACS over the visual cortex on the ability to terminate migraine attacks (2-hours post-stimulation) and pain due to migraines. Their results showed that pain was abated in 14 of 38 attacks in the active group compared to 0 of 23 in the sham group. Furthermore, they showed that self-reported pain was significantly less in active compared to sham group up to 4 hours post-stimulation.

The study is on an interesting topic with clear clinical applications that could benefit many people. The small sample size limits the application of the study’s findings, but I believe it still warrants publication after addressing some matters.

Authors: Thank you for your comments and suggestions.

Line 64 – 40 migraine patients estimation based on sample size calculations – was this calculated per group? Only 25 (16 real vs. 9 sham) total completed the study, so this would appear to be a severely under-powered study based on your sample size calculations.

Authors: It was calculated for the two groups, based on the sample size of previous brain stimulation studies, nevertheless, please note that at this stage we aimed to prove the feasibility of the methodology (15 patients/group would be suitable for this stage). The high drop out rate was a surprise for us, too. In the discussion it is mentioned, as one of the limitations, the high drop out rate and related small sample size.

Line 84 – has VAS shown to be valid and reliable for migraine pain? Please provide a reference

Authors: With regard to this point a correction was made, our scale was a numerical scale in the pain diary, therefore we replaced VAS and inserted NAS. The following sentence and reference was inserted into the section experimental design:

“NAS is frequently used as a valid and reliable measurement of migraine pain (Herd et al., 2019)”

Line 103-105 – since the stimulator was turned off after 30 seconds of stimulation, would the participants not be aware of this sudden offset of the machine rendering the “blinding” irrelevant?

Authors: In this study parallel group design was used, furthermore, the patients had no previous experience with regard to any kind of brain stimulation, therefore could not compare real and sham stimulation conditions (this information is stated in the Patients section). None of the patients mentioned the sudden offset of the stimulation during sham, probably because it was ramped down.

Lines 116-117 – Why was a Chi-squared test used for this analysis? Chi-squared test are used to show whether there is a relationship between two variables (without any direction of the relationship), and I don’t believe this is what you were trying to achieve with this analysis. An independent samples t-test or more likely Wilcoxon-Mann-Whitney (since it unlikely normally distributed data) should be used instead. Adjust results accordingly.

Authors: Thank you for this comment, you are right, the data were not normally distributed, we used a Mann Whitney U-test to compare the number of attacks, the level of significance did not changed.

Table 1- The sham group reported a lot more mean number of attacks per year – did you perform statistical analysis on this? If not, you should since it could be argued that the sham group suffered from a more serious migraines.

Authors: thank you for this comment. Yes, you are right, there were patients in the sham group with higher number of attacks, however, the variability in this group was high, therefore we found no statistical difference between groups.

“The demographical characteristics of the patients related to the disease (duration of the disease and number of attacks/year) did not differ significantly between the active and sham stimulation groups”.

Lines 137 – 138 – You should report average per person rather than total migraine attacks due to the uneven size of groups.

Authors: This information was inserted into the text, in both groups it was ~4 attacks/patient.

Lines 143 – 144 & Figure 2 – The active vs. sham looks different at 8 hours (more so than 0’); please confirm that pain was less in active at 0’ but not 8 hours.

Authors: The data ware reanalyzed and a new figure was made, with confidence intervals now. We are very sorry for the mistake.

Line 145 – spelling mistake - none not non

Authors: the sentence was corrected.

Table 2 – please fix font size and bolding.

Authors: the table was corrected.

Line 210 - Drop-out/exclusion rate was high suggesting at home application, although in essence potentially effective, does not necessarily ameliorate adherence to treatment due to barriers of attending a clinic. I think the claim that home therapy was well tolerated may be too general as those that didn’t use it and excluded from study/dropped-out may have not tolerated, hence not completed stimulation. Please reflect this in your discussion.

Authors: This sentence was rewritten, patients, who stimulated themselves, tolerated the stimulation well. Nevertheless, many of them who dropped out, did not even start the stimulation, therefore we cannot conclude that the drop out rate was related to non-tolerated adverse effects.

Figure 1 – Please define that VAS is used for pain severity in both title and y-axis.

Authors: The figure was corrected (now fig. 2).

Reviewer 2 Report

Dear editor,

thank you for giving me the opportunity to review this interesting work by Antal et al.

Overall, the results are interesting and the work well written. In particular, I find home treatment a very attractive application for TES, and the real frontier of non-invasive brain stimulation. However, there are some aspects that should be deepen and explained throughout the text:

  • The rationale for using tACS with 0.4 mA at 140 Hz at the level of the occipital cortex seems weak. Is there any evidence that this kind of stimulation may reduce occipital cortex excitability? (e.g. by evaluating changes in the phosphene threshold, in normal subjects or patients with migraine). Moreover, the choice of the cortical target (visual cortex) could be questionable in patients with migraine without aura. My concern is that the effect of tACS could be related to a more diffuse effect on brain functioning. This also considering that in the tACS protocol the reference electrode cannot be considered neutral. It would be interesting to investigate whether different electrode montages could result in similar or different clinical effects (e.g. targeting the motor cortex). These aspects could be discussed within the text especially to enrich discussion.
  • In the introduction (lines 41-44) the mechanism hypothesized to explain the therapeutic effect of tACS seems too generic and speculative.
  • If I haven't missed it, there is no definition of ‘terminated attack’, representing the primary endpoint.
  • A figure showing percentage of attacks effectively treated by tACS in the real and sham groups, also showing attacks requiring or not medication, should be included.
  • Were different effects observed in the treatment of attacks with and without aura? is there information on the effects of stimulation on the aura symptom? it would be very interesting to report such a finding, even if few patients could likely start the tACS treatment before the end of the aura symptoms.
  • Was there a correlation between efficacy of treatment and time interval between the onset of headache (or aura) and the start of tACS treatment?

Author Response

Reviewer 2.

Overall, the results are interesting and the work well written. In particular, I find home treatment a very attractive application for TES, and the real frontier of non-invasive brain stimulation. However, there are some aspects that should be deepen and explained throughout the text: The rationale for using tACS with 0.4 mA at 140 Hz at the level of the occipital cortex seems weak. Is there any evidence that this kind of stimulation may reduce occipital cortex excitability? (e.g. by evaluating changes in the phosphene threshold, in normal subjects or patients with migraine). Moreover, the choice of the cortical target (visual cortex) could be questionable in patients with migraine without aura. My concern is that the effect of tACS could be related to a more diffuse effect on brain functioning. This also considering that in the tACS protocol the reference electrode cannot be considered neutral. It would be interesting to investigate whether different electrode montages could result in similar or different clinical effects (e.g. targeting the motor cortex). These aspects could be discussed within the text especially to enrich discussion. In the introduction (lines 41-44) the mechanism hypothesized to explain the therapeutic effect of tACS seems too generic and speculative.

Authors: Thank you for your comments. Concerning the choice of the stimulation over the visual cortex, unfortunately there are no published data. Therefore we cannot provide better hypothesis. In the paper it is clearly stated that the present study a pilot, feasibility study. We agree, using different electrode montages (e.g. the return electrode over the primary motor cortex) might result in better clinical effects, nevertheless, according to our knowledge, these aspects were not investigated yet. The following sentence was inserted into the discussion:

“With regard to the stimulation montage, the placement of return electrode over the motor cortex that it was done in previous tDCS studies [46], and based on the fact that 140 Hz stimulation effectively modified the size of MEP amplitudes [16] might result in better clinical efficacy.”

With regard to the presence of the aura, rTMS and tDCS over the visual cortex were frequently applied not only patient with aura but also without aura. Unfortunately the number of the patients with aura/without aura in both groups is too low to make a conclusion with regard to this point.

If I haven't missed it, there is no definition of ‘terminated attack’, representing the primary endpoint.

Authors: Thank you for this comment, the definition was inserted (VAS < 1).

A figure showing percentage of attacks effectively treated by tACS in the real and sham groups, also showing attacks requiring or not medication, should be included.

Authors: The figure was inserted.

Were different effects observed in the treatment of attacks with and without aura? is there information on the effects of stimulation on the aura symptom? it would be very interesting to report such a finding, even if few patients could likely start the tACS treatment before the end of the aura symptoms.

Authors: No different effects were observed in patients with or without aura. There is no documented information on the aura symptoms in the patient diaries. This information is stated in the result section.

“With regard to the presence of aura, it was not possible to make reliable statistical analysis because of the low number of patients in each group, however, no different effects were observed in patients with or without aura.”

Was there a correlation between efficacy of treatment and time interval between the onset of headache (or aura) and the start of tACS treatment?

Authors: this time interval was not documented (or not precisely documented) in the migraine diary, therefore we could not make a correlation.
